# Spread of COVID-19 Infection in Long-Term Care Facilities of Trieste (Italy) during the Pre-Vaccination Era, Integrating Findings of 41 Forensic Autopsies with Geriatric Comorbidity Index as a Valid Option for the Assessment of Strength of Causation

**DOI:** 10.3390/vaccines10050774

**Published:** 2022-05-13

**Authors:** Martina Zanon, Michela Peruch, Monica Concato, Carlo Moreschi, Stefano Pizzolitto, Davide Radaelli, Stefano D’Errico

**Affiliations:** 1Department of Medical Surgical and Health Sciences, University of Trieste, 34149 Trieste, Italy; martina.zanon@virgilio.it (M.Z.); michela.peruch95@gmail.com (M.P.); monica.concato@studenti.units.it (M.C.); davide_radaelli@hotmail.it (D.R.); 2Department of Medicine, Forensic Medicine University of Udine, 33100 Udine, Italy; carlo.moreschi@uniud.it; 3Department of Pathology, Santa Maria della Misericordia University Hospital, 33100 Udine, Italy; stefano.pizzolitto@asufc.sanita.fvg.it

**Keywords:** COVID-19, cause of death, autopsy, strength of causation, Geriatric Index of Comorbidity, frailty scores, long care term facilities, older people

## Abstract

Background: in 2020, a new form of coronavirus spread around the world starting from China. The older people were the population most affected by the virus worldwide, in particular in Italy where more than 90% of deaths were people over 65 years. In these people, the definition of the cause of death is tricky due to the presence of numerous comorbidities. Objective: to determine whether COVID-19 was the cause of death in a series of older adults residents of nursing care homes. Methods: 41 autopsies were performed from May to June 2020. External examination, swabs, and macroscopic and microscopic examination were performed. Results: the case series consisted of nursing home guests; 15 men and 26 women, with a mean age of 87 years. The average number of comorbidities was 4. Based only on the autopsy results, the defined cause of death was acute respiratory failure due to diffuse alveolar damage (8%) or (31%) bronchopneumonia with one or more positive swabs for SARS-CoV-2. Acute cardiac failure with one or more positive swabs for SARS-CoV-2 was indicated as the cause of death in in symptomatic (37%) and asymptomatic (10%) patients. Few patients died for septic shock (three cases), malignant neoplastic diseases (two cases), and massive digestive bleeding (one case). Conclusions: Data from post-mortem investigation were integrated with previously generated Geriatric Index of Comorbidity (GIC), resulting in four different degrees of probabilities: high (12%), intermediate (10%), low (59%), and none (19%), which define the level of strength of causation and the role of COVID-19 disease in determining death.

## 1. Introduction

In the first 5 months of the SARS-CoV-2 pandemic in Italy, 32,981 cases of COVID-19-related deaths were reported, with a mean age of 81 years and only 1% occurring in people under 50 years [1,2]. A similar trend was observed in USA, China, and northern Europe, thus providing important information regarding the increased risk of mortality in the higher age groups [3,4,5,6]. It was observed that individuals >59 years of age are five times more likely to die following the onset of COVID-19 symptoms as compared to those between the ages of 30–59 [7,8]. Long-term care facilities were dramatically hit by the COVID-19 outbreak, where a death percentage of 25–85% was recorded worldwide [9,10,11,12]. It has been observed that long-term care facilities consisted of a high-risk population in a high risk setting (Gardner) [13,14,15,16]. Many authors described an over 50 times higher death rate in residents of long-term care facilities than community-dwelling older people [17,18,19,20,21]. It was observed that characteristics of the residents in long-term care facilities (age, higher body mass index, male sex, and renal impairment) had a different impact on the death rate as well as the quality level of healthcare and the adequacy of safety measures adopted to face the spread of the infection and its deadly consequences (specific training of the staff, good staff/resident ratio) [21,22,23,24,25]. In Italy, the territorial distribution of long-term care facilities (or residential care homes) exhibits a gradient moving from the North to the South of the country and has been considered significant in the distribution of death during the COVID-19 pandemic [26]. Concerns about the possibility that personnel represented the source of COVID-19 introduction and promoted the spread of the infection among the residents favored the onset of claims and triggered efforts to ensure equal access to high-quality healthcare across long-term care facilities as well as to provide the extensive vaccination of residents and personnel [27,28,29]. The Italian health authorities, for example, promoted COVID-19 vaccination of the residents of long-term care facilities, their relatives, and healthcare personnel as a priority in the vaccination plan, with the aim to reduce the risk of severe COVID-19 disease in older people and the risk of hospitalization, to guarantee the continuity of care [30,31]. In May 2021, the Italian health authorities published a survey of the spread of COVID-19 infection in about 41% of all Italian long-term care facilities; of the 9154 patients who died, 680 had a positive nasopharyngeal swab and 3092 had flu-like symptoms, and a death rate of 0.7 per 100 residents was estimated, 7.4% of which involved residents with SARS-CoV-2 infection [32]. A 3:1 ratio of mortality was estimated when comparing the long-term care facilities’ residents and people aged over 70 living in the community. There can be no doubt that comorbidities in older people affected by COVID-19 infection played a determinant role in their death [33]. Mehra MR et al. showed that underlying cardiovascular disease and chronic obstructive pulmonary disease were associated with a higher mortality rate amongst hospitalized COVID-19 patients [34]. Also, Lippi G et al. observed a five and half times higher risk of developing a severe infection due to SARS-CoV-2 in older people affected by COPD [35]. The distinction between “died from” and “died with” COVID-19 remains a topic of debate especially in older people where the role of comorbidities needs to be accurately assessed [36,37]. The international guidelines for certification and classification of COVID-19 as the cause of death defined a COVID-19-related death on the basis of a clinically compatible illness, in a probable or confirmed COVID-19 case, unless there is a clear alternative cause of death that cannot be related to COVID-19 disease [38]. The Italian health authorities updated the definition of death due to COVID-19 infection based on objective criteria, although in the same period they discouraged autopsies in suspected or confirmed cases of COVID-19 (Table 1) [39,40,41,42,43,44,45,46].

Some authors proposed different standardized models to better define the cause of death in older people with COVID-19 infection on the basis of clinical data or autopsy findings, respectively [47,48,49,50,51].

The purpose of this study is to validate a methodological approach in the assessment of the strength of causation between COVID-19 infection and death, by integrating data collected from 41 consecutive autopsies of older people residents in long-term care facilities who died in the period May–June 2021 with Geriatric Index of Comorbidity (GIC), commonly used in clinical practice as a measure of frailty.

## 2. Materials and Methods

Between the beginning of May and the beginning of June 2020, a total of 41 consecutive deaths occurred in five different long-term care facilities where the spread of COVID-19 infection was reported. A complete post-mortem examination was performed in each case according to existing safety measures and protocols in clinical and forensic autopsies [43,52,53,54,55]. Before and during autopsies, multiple swabs were collected at identical sites from rhynopharinx, oropharynx, trachea, right and left bronchus, and rectum, which were examined by Real Time Polymerase Chain Reaction (RT-PCR). Positive results were reported as semi-quantitative Cycle Threshold (Ct) values [56]. The Ct values, collected and grouped into three categories, were defined as follows: <25 strongly positive, 25–35 moderately positive, and >35 weakly positive [57]. Gross examination of the head, thoracic, and abdominal organs was performed in each case, followed by extended sampling of organs for histological examinations with routine hematoxylin and eosin staining. Histological examination was integrated with immunohistochemistry by means of anti-core and anti-spike proteins and anti-CD4-8 panel antibodies in cases of interstitial lymphocytic infiltration. Data regarding age, sex, comorbidities, onset of symptoms before death, and the results of swabs when performed before the death were extracted from medical records and collected in an extended database. The level of frailty was estimated for each resident according to the Geriatric Index of Comorbidity (class I to IV) on the grounds of the number of reported diseases and the severity of the diseases measured by the Greenfield’s Individual Disease Severity (IDS) by grading each condition on a 0–4 scale on the basis of the following general framework: 0 = absence of disease, 1 = asymptomatic disease, 2 =symptomatic disease requiring medication but under satisfactory control, 3 = symptomatic disease uncontrolled by therapy, and 4 = life-threatening disease or greatest severity of the disease [58]. Class I includes residents with one or more conditions with IDS = 1, class II included patients with one or more conditions with IDS = 2, class III includes patients with one condition with IDS = 3, and class IV included residents with two or more conditions with IDS = 3 or one or more conditions with IDS = 4 [59].

Data from post-mortem investigation were integrated with the Geriatric Index of Comorbidity (GIC), resulting in four different strengths of causation (high, intermediate, low, and none), which define the level of strength of causation and the role of COVID-19 disease in determining death.

## 3. Results

*Description of the population.* Forty one consecutive deaths occurred among the residents of five different long term care facilities from May to June 2020. The mean age was 87 (range 57–99) with a prevalence of female sex (63%). Sixty one percent of residents were symptomatic within 10 days before the death, suffering from fever (44%), dyspnea or respiratory failure in therapy with oxygen (39%), drowsiness (12%), hypotension (12%), and acute renal failure (7%). In 45% of cases, death occurred unexpectedly. In 80% of symptomatic residents, an oropharyngeal swab was performed before death, 72% of which showed a positive result for COVID-19.

*Geriatric Index of Comorbidity*. The number of comorbidities varied from 0 to more than 7, with an average number of 4. The most frequent was severe walking impairment/bed rest syndrome (73%) followed by severe cognitive impairment (61%), hypertension (41%), ischemic heart disease (34%), COPD (29%), chronic kidney failure (29%), arrhythmogenic heart disease (24%), neoplastic disease with dissemination (17%), congestive heart failure/valvulopathy (17%), and diabetes mellitus (12%). The severity of frailty was measured for each resident based on the Geriatric Index of Comorbidity (GIC). None of the deceased belonged to class I, 2% of the cases belonged to class II, 20% of the cases belonged to class III, and 78% of the cases belonged to class IV (Table 2).

*Post mortem swabs*. The mean time between death and swab collection was 36 days (range 12–60). Swabs were analyzed by real time PCR and in 34 cases (83%) one or more of these resulted positive for COVID-19. The range of cycle threshold for the nasopharyngeal swabs was 15.47–37.11 cycles, for the oropharyngeal swabs 17.64–36.35 cycles, for the tracheal swabs 18.08–35.45 cycles, for the right bronchus swabs 19.69–36.7 cycles, for the left bronchus swabs 16.65–36.56 cycles, and for the rectal swabs 32.62–35–72 cycles. Based on Ct values, 19% of the swabs resulted highly positive (Ct < 24), 27% moderately positive (Ct 24–33,99), and 7% weakly positive (Ct < 35) (Table 3).

*Autopsy findings*. The mean time between death and autopsy was 36 days (range 12–60). All cadavers were preserved in cold or refrigerated rooms before autopsy. Heavy, congested, and edematous lungs were reported in all autopsies. Lung cancer with liver metastasis was observed in only one case. Pleural fluid was reported in only two cases as well as pleural adhesions. The mean weight of the right and left lung was 621 gr (range 260–1000) and 559 gr (range 180–940), respectively. The mean weight of the brain was 1256 gr (range 880–1700), that of the heart was 407 gr (range 240–700), that of the liver was 912 gr (range 560–1440), that of the spleen was 125 gr (range 40–360), and that of the right and left kidney was 112 gr (range 40–350) and 112 gr (range 40–350), respectively. A massive pulmonary embolism was recorded in only one case as well as purulent peritonitis and gastrointestinal hemorrhage due to gastric ulcer. Microscopic examination was heavily influenced by the advancing of putrefactive changes. Vascular changes associated with proliferative and exudative diffuse alveolar damage with hyaline membrane deposition, necrosis of alveolar lining cells, type II pneumocyte hyperplasia with nucleomegaly associated with the accumulation of macrophages, and multinucleated giant cells were observed in 24% of cases. Mild interstitial focal infiltration with peribronchiolar and perivascular CD-3 positive T cells with a predominance of CD4-positive T cells over CD8-positive T cells was also described in 48% of cases. Fibrin thrombi in pre-capillary and post-capillary vessels were observed in lung specimens in 15% of residents. Findings of chronic obstructive pulmonary disease were observed in 73% of cases. In 66% of cases, features of bacterial pneumonia were described.

In 73% of cases, pathologic cardiac features were generally related to chronic cardiovascular comorbidities (ischemic dilated cardiomyopathy and myocardial scarring), and in 54% of cases severe coronary artery disease was recorded. Advanced putrefactive changes limited histopathological observation of renal samples so that acute kidney injury (AKI) could not be excluded at all. In 73% of cases, arterionephrosclerosis and pre-existing pathological features of hypertension and diabetes were observed as well as focal and sparse chronic inflammatory infiltrates. On the basis of autopsy findings, acute respiratory failure due to bacterial bronchopneumonia was indicated as the cause of death in 31% of residents. Eight percent of residents with one or more positive swabs for SARS-CoV-2 died as a result of the diffuse alveolar damage. Acute cardiac failure in residents affected with chronic ischemic cardiomyopathy was indicated as the cause of death in 37% of residents with one or more positive swabs for COVID-19. Acute cardiac failure in residents affected with chronic ischemic cardiomyopathy without signs of COVID-19 was indicated as the cause of death in 10% of residents. In 8% of residents, death was related to septic shock and multiple organ failure syndrome. Four percent of the deceased died from advanced malignant disease (stage IV) complicated by bacterial bronchopneumonia. In one case, the death was related to acute hemorrhagic shock due to massive digestive bleeding.

*Degree of strength of causation.* Data from post-mortem investigation were integrated with previously generated Geriatric Index of Comorbidity (GIC), resulting in four different degrees of probabilities: high (12%), intermediate (10%), low (59%), and none (19%), which define the level of strength of causation and the role of COVID-19 disease in determining death (Table 4).

## 4. Discussion

Deaths related to COVID-19 as a direct or indirect result of SARS-CoV-2 infection and correct attribution to the pandemic may be a challenge, especially in older people. In this population, heavily affected during the first wave of the pandemic, the higher vulnerability and the presence of multiple comorbidities contributed significantly to an overestimation of the phenomenon, also because of the lack of a systematic post-mortem examination. Dying from COVID-19 or with COVID-19 represented an unsolved dilemma in residents of long-term care facilities, where clinical manifestations of COVID-19 may be difficult to recognize because typical symptoms such as fever, cough, and dyspnea may already be present due to other comorbidities like several pulmonary diseases or may also have specific or atypical presentations like anorexia, diarrhea, fatigue, headache, and dizziness, and clinical data from medical records are scarce, when available [2,60,61,62]. In this scenario of uncertainty, Italian long-term care facilities are going through a judicial pandemic because of the spread of deadly COVID-19 infections among residents.

In our study, a methodological approach is proposed for the study of COVID-19-related death in a selected population of 41 older residents who died in the bimester May–June 2020 in five different long-term care facilities with suspected or confirmed COVID-19 infection. The data provided by complete autopsies were integrated with the severity of frailty measured by the Geriatric Index of Comorbidity for the purpose of assessing the role of comorbidities in the mechanism of death and to provide an affordable definition of the causes of death based on objective criteria.

The small sample size is unlikely to be fully representative of older people, however, it provides an estimate of the true number of mortalities directly related to the pandemic in older people affected by multiple comorbidities in community contexts.

The characteristics of the population observed differ, in fact, from other studies, because of the peculiarity of the context (long-term care facilities) with a prevalence for the female sex (63%) and mean age of 87 years, higher than that of hospitalized patients. In the biggest autopsy studies in the literature, mean age was between 69 and 79 years, with a prevalence of male patients (58–87%) [48,50,63,64,65,66,67]. Only two case series of elder patients or patients who died in community settings had a prevalence of female patients (55–59%), with a mean age of 88 and 72 years [49,68].

The number of comorbidities varied from one to more than seven, with 85% of the subjects affected by more than three comorbidities. Hypertension, complications of diabetes mellitus, and chronic obstructive pulmonary disease were most frequently reported as expected in consideration of the data observed in other studies, where diabetes was seen in 8–44% of the cases, hypertension in 22–100% of the deceased, and chronic obstructive pulmonary disease in 6–55% of the patients who underwent an autopsy [43,48,63,64,65,66,67,68,69]. The severe walking impairment/bed rest syndrome and the severe cognitive impairment observed in most cases characterized our population and can be considered as an independent prognostic factor for death, highly impacting on the severity of frailty [44,70]; in fact, walking impairment was not identified in any of the previous studies, and chronic neurological conditions were seen in 10–41% of the cases [43,48,63,64,65,66,68,69]. As observed by other authors, fever, dyspnea, and respiratory failure were described in the days prior to death in 55–68%% of cases [49,65,68]. In our case series, the presence of fewer symptomatic patients could probably be influenced by the scarcity of data reported in the medical records of the residents of long-term care facilities.

Autopsy macroscopic findings were compatible with what is reported in other studies, with a mainly pulmonary engagement with increased weight, congestion, and oedema, which were found in 78–100% of the cases [49,63,66,69], and histological features of exudative and proliferative diffuse alveolar damage found in 67–100% of the deceased [48,49,63,66,67,69]. Unlike data from literature where pulmonary thromboembolism was found in 6–25% of patients [43,50,66,67,69], only a small number of patients of our case series showed microthrombi in alveolar capillaries or in small vessels, probably due to the limited time between infection and death as Youd and Moore pointed out [50]. Advanced putrefactive phenomena limited the recording of acute kidney injury.

Significant results were provided by the analysis of cycle threshold of multiple swabs performed before and during autopsy. The Ct value is defined as the number of cycles of amplification required for the fluorescent signal to cross the threshold, which is above the background signal. Therefore, Ct values are inversely proportional to the amount of target nuclei acid present in the tested sample. Recent studies demonstrated that qRT-PCR Ct values correlate strongly with the cultivable virus, providing a valuable surrogate for infectious virus detection in biological samples [71,72].

In our study, the mean time between death and swab collection was 36 days (range 12–60) more than in other studies in which the persistence of viral RNA was observed in post-mortem collected swabs up to 128 h after death or in buried and exhumed corpses [73,74,75,76]. Swabs were analyzed by real time PCR, 42% of which were from moderate (27%) to highly positive (15%).

The parameters of vulnerability were chosen among the several scoring systems available in clinical practice [77]. We decided to integrate data deriving from autopsies with the GIC considered to be the most accurate predictor of death to evaluate the impact and burden of the diseases on the mechanism of death [78]. It enabled the grouping of residents into four categories characterized by a different strength of causation, and in 78% of cases the role of COVID-19 in determining death was excluded.

## 5. Conclusions

The current classification of COVID-19-related deaths may be ineffective, and the risk of an overestimation of mortality rate is realistic in this selected population, especially if a complete autopsy is not performed systematically. The integration of autopsy findings and the assessment of severity of comorbidities by means of frailty scores used in clinical practice has proved to be a valid option for assessing the strength of causation. The use of this index currently represents a diagnostic proposal to identify whether SARS-CoV-2 infection represents the cause of death in subjects with multiple comorbidities. However, this methodology should be used on larger case studies to validate its use in assessing the cause of death.

## Figures and Tables

**Table 1 vaccines-10-00774-t001:** Definition of COVID-19-related death according to the Italian Istituto Superiore di Sanità (ISS).

Strenght of Causation	Criteria #1	Criteria #2	Criteria #3	Criteria #4
Certain	The death occurred in a COVID-19 case (confirmed)	Clinical (fever, cough, dyspnoea, dizziness, etc.) and radiological features of COVID-19 infection	No other plausible cause of death other than COVID-19 infection	Lack of wellbeing among COVID-19 infection and the death
Probable	The death occurred in a COVID-19 case (probable)	Clinical (fever, cough, dyspnoea, dizziness, etc.) and radiological features of COVID-19 infection	No other plausible cause of death other than COVID-19 infection	Lack of wellbeing among COVID-19 infection and the death
Possible (suspect)	The death occurred in a COVID-19 case (suspect)	Clinical (fever, cough, dyspnoea, dizziness, etc.) and radiological features of COVID-19 infection	No other plausible cause of death other than COVID-19 infection	Lack of wellbeing among COVID-19 infection and the death

**Table 2 vaccines-10-00774-t002:** Number of comorbidities and Geriatric Index of Comorbidity (GIC).

Comorbidities (n)	Residents (%)
*0–2* *3–4* *5–6* *>7*	15 44 39 2
**Geriatric Index of Comorbidity (GIC)**	
*Class I*	0
*Class II*	2
*Class III*	20
*Class IV*	78

**Table 3 vaccines-10-00774-t003:** Distribution of swab results and Ct range.

	Nasopharingeal	Oropharingeal	Tracheal	Right Bronchus	Left Bronchus	Rectal
Positive (n/%)	24/58.5%	25/61%	26/63.4%	28/68.3%	25/61%	4/10%
Negative (n/%)	17/41.5%	16/39%	15/36.6%	13/31.7%	16/39%	37/90%
Ct range	15.47–37.11	17.64–36.35	18.08–35.45	19.69–36.7	16.65–36.56	32.62–35.72
Ct < 23,99 (highly positive)	9	14	8	7	9	0
Ct 24–33,99 (moderately positive)	14	9	16	16	11	2
Ct > 34 (weakly positive)	1	2	2	5	5	2

**Table 4 vaccines-10-00774-t004:** Strength of causation between COVID-19 infection and death.

Strength of Causation of COVID-19 Infection	Characteristics	%
High (relevant to death)	≥1 positive post mortem swab, signs of diffuse alveolar damage (DAD), with or without bacterial pneumonia, regardless of GIC class	12%
Intermediate (contributing to death)	≥1 positive post mortem swab, moderate signs of COVID-19 infection (e.g., interstitial lymphocytic infiltrates, type II pneumocyte hyperplasia, microthrombosis) with bacterial pneumonia and GIC class < IV	10%
Low (weakly related with death)	≥1 positive post mortem swab, moderate or no signs of COVID-19 infection (e.g., interstitial lymphocytic infiltrates, type II pneumocyte hyperplasia, microthrombosis), with or without bacterial pneumonia and GIC class IV	59%
None (unrelated with death)	certain alternative cause of death, regardless of the outcome of the swab	19%

## Data Availability

All the data are in the hands of the authors and can be shown if requested.

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
