# Peer review of "Spread of COVID-19 Infection in Long-Term Care Facilities of Trieste (Italy) during the Pre-Vaccination Era, Integrating Findings of 41 Forensic Autopsies with Geriatric Comorbidity Index as a Valid Option for the Assessment of Strength of Causation"

_vaccines, 2022, doi:10.3390/vaccines10050774_

Round 1

Reviewer 1 Report

The manuscript is worth considering. It concerns a very intriguing issue, namely what was a true direct cause of death in very old patients infected with SARS-CoV-2 but simultaneously affected by multimorbidity and presenting physiologically attenuated immune system. The conclusion is of great interest.

Abstract: line 20-21: the sentence is obscure, needs rephrasing: "the defined cause of death was respiratory failure in subjects histological signs of diffuse alveolar damage in 8% of the cases". The same in line 23: "10% died of acute cardiac failure in patients without signs of Covid-19".

Discussion: Too many own data repeated from the section "Results" contrary to too few references to the other authors.

Conclusion: The sentence "Due to their fragility and context, older people residents in long-term care facilities are at higher risk of death in case of Covid-19 infection." should be removed as too obvious and not relevant to  the study.

Reviewer 2 Report

Comments to Authors:

Zanon et al. showed the geriatric comorbidity index as valid measurment in forensic autopsies of covid-19 care facilities in  italy. Overall, the manuscript may bring more interest to know the covid-19 related fingings, however, the manuscript still need huge correction, polish the manuscript in publication standard!

Major Comments:

1. Title: Title is too lengthy and confusing using too techincal terms(eg geriatric comorbidity)??? It is difficult to understand the readers please repharse concise!
2. Abstract:  The abstract need to revise carefully free from errors! Please see the sentence on the abstract! e.g. "Objective: to determine  whether Covid-19 was the cause of death in a series of older adults residents of nursing care home".

3. The sampleing you showed 41 for only May-June 2020? Why dont show 
other months cases? The sample numbers was poor, you need to inclusion of more sampling for statstics?

4. Results: Results need to correlate with other studies and concluded the each results sections?
5.  Conclusion: You should mention future direction of this study?

Round 2

Reviewer 2 Report

Revised manuscript may be acceptable